# Neural Architecture Search for Visual Anomaly Segmentation

**Tommie Kerssies**[1]  **Joaquin Vanschoren**[2]

[1,2]Eindhoven University of Technology

**Abstract**  This paper presents the first application of neural architecture search to the complex task of segmenting visual anomalies. Measurement of anomaly segmentation performance is challenging due to imbalanced anomaly pixels, varying region areas, and various types of anomalies. First, the region-weighted Average Precision (rwAP) metric is proposed as an alternative to existing metrics, which does not need to be limited to a specific maximum false positive rate. Second, the AutoPatch neural architecture search method is proposed, which enables efficient segmentation of visual anomalies without any training. By leveraging a pre-trained supernet, a black-box optimization algorithm can directly minimize computational complexity and maximize performance on a small validation set of anomalous examples. Finally, compelling results are presented on the widely studied MVTec dataset, demonstrating that AutoPatch outperforms the current state-of-the-art with lower computational complexity, using only one example per type of anomaly. The results highlight the potential of automated machine learning to optimize throughput in industrial quality control. The code for AutoPatch is available at: https://github.com/tommiekerssies/AutoPatch.

## 1 Introduction

The widespread adoption of computer vision for industrial quality control has underscored the importance of effective anomaly segmentation methods. Conventional methods necessitate a labeled dataset comprising both normal and anomalous examples. However, in many industries, the occurrence of anomalies is rare. Consequently, collecting a large dataset of anomalies presents a considerable challenge.

A variety of unsupervised anomaly segmentation methods have been proposed that circumvent the need for anomalies during the training process. These methods are predicated on the notion that anomalies constitute rare events that deviate markedly from normal patterns. As such, these anomalies can be segmented by identifying image regions with low probability under a probabilistic model of normal data. Despite the encouraging performance of deep feature embedding based methods for unsupervised anomaly segmentation [13], prior work has largely overlooked the significance of computational complexity. In the context of industrial applications, reducing the computational complexity is crucial to optimize throughput.

Neural architecture search has been proposed to automate the design of more efficient neural architectures, where the goal is to discover Pareto optimal architectures that simultaneously achieve high performance while minimizing computational complexity. A supernet shares weights among its subnets to reduce computational cost and accelerate the search for neural architectures [5]. Although weight-sharing alleviates the burden of training each possible architecture from scratch, implementing such a supernet for a task-specific neural architecture design space and subsequently training it on domain-specific data is challenging [10].

Nearest-neighbor scoring has recently gained popularity as an effective approach to unsupervised anomaly segmentation [12, 2, 14, 7, 6]. These methods do not rely on fine-tuning, but instead utilize a frozen pre-trained encoder (e.g., on ImageNet [8]) to encode images into patches. This approach is well-suited to neural architecture search, as subnets from a pre-trained supernet can directly be used as encoder without any additional training.

Prior work chooses the encoder architecture and its extraction layers and patch sizes based on empirical observations on the test set of anomalies [12]. However, different data distributions show different optima, due to inherent trade-offs between highly localized predictions, a more global context, and bias in deeper layers from the pre-training task of ImageNet classification [12]. Searching for the encoder architecture, extraction layers, and patch sizes by hand is not trivial and may lead to suboptimal results. Furthermore, a fixed neural architecture is not most efficient. Therefore, this paper introduces the AutoPatch method to automatically search for Pareto optimal neural architectures and their extraction layers and patch sizes.

In the scenario of insufficient anomaly data to train a model by conventional supervision, the challenge in applying neural architecture search to unsupervised anomaly segmentation is that there may only be a few anomalous examples to evaluate the performance of the architectures in the search space. Therefore, the goal of this paper is to investigate the relationship between the number of anomalies used for the search and the generalization of the discovered architectures to unseen anomalies, leading to the following research question: *"To what extent can the Pareto optimal neural architectures for visual anomaly segmentation be approximated, given limited anomalies?"*

The main contributions of this paper are as follows: (1) proposal of the region-weighted Average Precision (rwAP) metric for measuring anomaly segmentation performance, (2) introduction of the AutoPatch neural architecture search method, which enables efficient segmentation of anomalies without any training, and (3) demonstration that AutoPatch does not require many anomalous examples.

## 2 Background and Related Work

**Visual anomaly segmentation** is the focus of this paper, which can be defined as a binary pixel-wise classification task. The MVTec dataset [3] is introduced as a dataset to benchmark unsupervised anomaly detection and segmentation methods. The dataset comprises five texture and ten object categories, with normal images for training and both normal and anomalous images for testing. Anomalies are grouped by type (e.g., dent, scratch), and binary pixel-wise annotations are provided for evaluation.

Anomaly segmentation methods generate pixel-wise anomaly scores, which require an anomaly score threshold to classify a pixel as anomalous. The optimal threshold depends on the desired trade-off between detecting all anomalies (high recall) and minimizing false positives (high precision). Determining this threshold without (many) anomalies can be challenging [3]. Threshold-free metrics, such as area under the receiver operating characteristic curve (AUROC) and average precision (AP), assess performance without needing a threshold. AUROC measures the trade-off between false positive rate (FPR) and true positive rate (TPR/recall) by computing the area under the receiver operating characteristic (ROC) curve, which plots FPR versus TPR (recall). AP measures the average precision at each recall (TPR) level by computing the area under the precision-recall (PR) curve, which plots precision versus recall (TPR).

When treating each pixel individually, different sized regions are not taken into account. Consequently, more importance is given to large anomalous regions (more pixels), while small anomalous regions (fewer pixels) have less influence on the metric. In practice, smaller anomalous regions are often equally as important as larger ones. To address this, the per-region overlap metric (PRO) is proposed [3], calculated by weighting the TPR over each ground truth region instead of all pixels. Similarly to AUROC, the area under the PRO curve (AUPRO) is calculated as the area under the curve plotting FPR versus PRO.

The MVTec dataset [3] has a large class imbalance, with only 2.7% of the pixels in the test set labeled anomalous. This causes AUROC and AUPRO to produce overly optimistic estimates of performance, as the number of true negatives is usually much higher than the number of false positives. In other words, the segmentation results are only meaningful at a relatively low FPR. To account for this, the authors of MVTec [3] suggest computing the normalized area under the ROC

or PRO curve to a specific maximum FPR of 0.3. When varying this limit, the authors find that the ranking of some anomaly segmentation methods changes significantly, therefore, they conclude that the limit needs to be carefully set depending on the requirements of the application.

**Nearest-neighbor scoring** has recently gained popularity as an approach to unsupervised anomaly segmentation [12, 2, 14, 7, 6]. Such methods are not based on supervised learning, but instead use a frozen pre-trained encoder (for example, on ImageNet [8]) to encode images. During "training", the features of normal images are stored in a memory bank. During inference, the features of an image are used to find their nearest neighbors in the memory bank. The nearest neighbor distances are used as anomaly scores.

The recently introduced method PatchCore [12] shows promising anomaly segmentation performance on the challenging MVTec dataset [3]. The method extracts patches from an image by aggregating features from multiple mid-level layers of the encoder. To account for sufficient anomalous context, robust to local spatial variations, feature aggregation is performed over the local neighborhoods of patches. To speed up the nearest-neighbor scoring, the authors introduce a subsampling algorithm to reduce the size of the memory bank to a *coreset*, by only keeping the most informative patches and thereby reducing redundancy. As a consequence, the encoder becomes relatively more important in its contribution to the total computational complexity. Finally, the WideResNet-50 [15] encoder used by default may not be suitable for deployment on edge devices with limited resources.

**Multi-objective black-box optimization** algorithms are commonly used for hyperparameter optimization, as hyperparameters cannot be directly optimized by gradient descent. Multi-objective optimization algorithms aim to find Pareto optimal trade-offs between (conflicting) objectives, such as performance and computational complexity in the context of neural architecture search. The Multi-Objective Tree Parzen Estimator (MOTPE) [11] is an effective algorithm for multi-objective black-box optimization. MOTPE is an extension of the Tree-structured Parzen Estimator (TPE) [4], which models the conditional probability of hyperparameters given the performance of previously evaluated configurations. MOTPE adapts the TPE to handle multiple objectives simultaneously. The multi-variate variant [1] of the MOTPE algorithm models the multi-variate dependencies between parameters explicitly, instead of sampling them independently.

**Weight-sharing neural architecture search** methods have emerged as an efficient way to find optimal neural network architectures. In the Once-for-All [5] framework, a supernet is designed consisting of multiple subnets that share weights. The supernet is trained only once, allowing for the extraction of specialized subnetworks that can be deployed under different domains, tasks, hardware and latency constraints. This approach to neural architecture search significantly reduces the search cost and computational resources required to find optimal architectures, compared to neural architecture search methods that train each candidate architecture independently.

## 3 Region-weighted Average Precision

To assess the performance of anomaly segmentation, prior work [3] suggests limiting the area under the ROC and PRO curve to a specific maximum FPR. However, these metrics lack flexibility for diverse requirements and could potentially lead to inaccurate performance indications in the absence of an appropriate FPR limit [3]. Consequently, this paper advocates the adoption of the Average Precision (AP) metric, which circumvents the need for FPR limit adjustments due to its independence from FPR in its calculation, thereby addressing the issue of large class imbalance.

In this context, the region-weighted Average Precision (rwAP) metric is proposed, designed to better align with real-world performance. To compute the rwAP, the PR curve is multiplied by a weighting factor for each region. This weighting factor depends on both the size of the region and

the number of regions that represent the same type of anomaly. This approach ensures that regions of varying sizes and types of anomalies with distinct region counts receive equal importance in the evaluation process. The weighting factor for region $j$ in image $i$ of type $k$ is defined as:

$$w_{i,j} = \frac{A_{mean}}{A_{i,j}} \times \frac{N_{mean}}{N_k} \tag{1}$$

where $A_{mean}$ is the mean area for all regions, $A_{i,j}$ represents the area of region $j$ in image $i$, $N_{mean}$ signifies the mean number of regions across all types, and $N_k$ is the count of regions of type $k$. The region-weighted true positives and false negatives are computed as:

$$\text{rwTP}(t) = \sum_{i,j} w_{i,j} \times \text{TP}(i, j, t) \qquad \text{rwFN}(t) = \sum_{i,j} w_{i,j} \times \text{FN}(i, j, t) \tag{2}$$

where $\text{TP/FN}(i, j, t)$ corresponds to the number of true positive / false negative pixels for region $j$ in image $i$ at threshold $t$, while $w_{i,j}$ denotes the previously defined weighting factor for the respective region. Subsequently, the region-weighted PR curve is computed as:

$$\text{rwP}(t) = \frac{\text{rwTP}(t)}{\text{rwTP}(t) + \text{FP}(t)} \qquad \text{rwR}(t) = \frac{\text{rwTP}(t)}{\text{rwTP}(t) + \text{rwFN}(t)} \tag{3}$$

where FP denotes the number of false positives at threshold $t$, which cannot be associated with a region and thus remains unweighted. Analogously to the AP score, the rwAP score is computed as:

$$\text{rwAP} = -\sum_{i=1}^{N-1} (\text{wR}_i - \text{wR}_{i-1}) \times \text{wP}_i \tag{4}$$

where $N$ refers to the number of thresholds on the PR curve.

## 4 Methodology

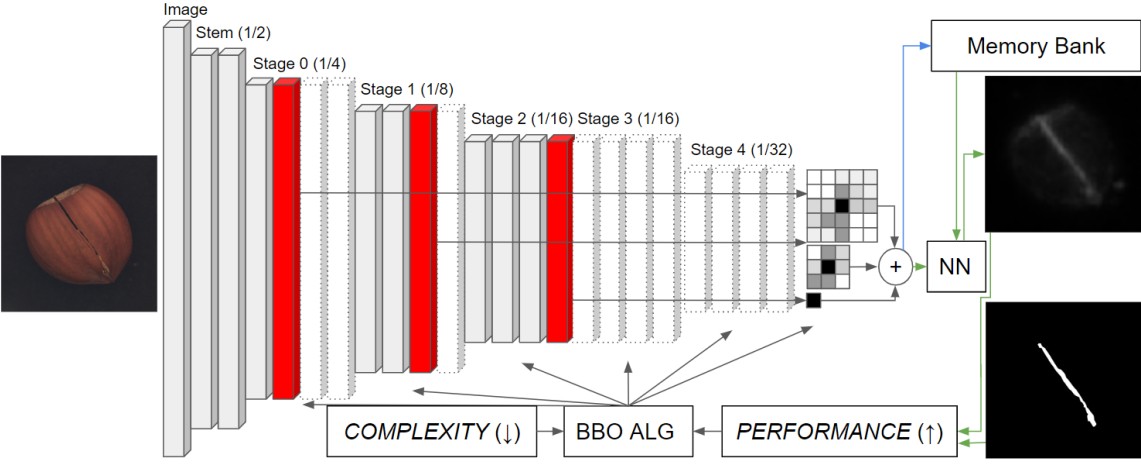

Figure 1: Overview of the AutoPatch method. The blue line pertains exclusively to training, while the green lines are specific to inference. *NN* denotes the nearest-neighbor scoring. *BBO ALG* refers to the black-box optimization algorithm.

Drawing inspiration from PatchCore [12], this paper proposes a simple anomaly segmentation process. Given an encoder and the layers designated for extraction, an image is passed through the

encoder to the final extraction layer. Average pooling with a variable kernel size (patch size) and a stride of 1 is applied to the layer outputs. The output of the deeper layers undergoes upsampling with bilinear interpolation to match the output resolution of the first extracted layer. The feature maps from distinct layers are then concatenated to form the final patches. During "training", a flat list of patches from normal training images is stored in the memory bank. During inference, patches of an image are used to find their nearest neighbors in the memory bank. The lower resolution patch distances are upsampled via bilinear interpolation to the original image resolution. The pseudo-code for inference is provided in Algorithm 1.

---

**Algorithm 1** Inference for the simple anomaly segmentation process, inspired by PatchCore [12].

---

**Require:** input image $X$, encoder $\text{Encoder}(\cdot)$, memory bank $M$, patch sizes $P$, extraction layers $E$
1: $Z \leftarrow \text{Encoder}(X)$
2: Initialize empty list $Patches$
3: **for** $i = 0$ to $\text{len}(Z)-1$ **do**
4:     **if** $i \in E$ **then**
5:         $Z_i \leftarrow \text{AvgPool2D}(Z_i, kernel\_size = P_i, stride = 1)$
6:         $Z_i \leftarrow \text{Interpolate}(Z_i, Z_0.resolution)$
7:         Add $Z_i$ to $Patches$
8:     **end if**
9: **end for**
10: $Patches \leftarrow \text{Concatenate}(Patches, axis = 1)$     ▷ Concatenate along the channel dimension
11: $\hat{Y} \leftarrow M.\text{Search}(Patches)$
12: $\hat{Y} \leftarrow \text{Interpolate}(\hat{Y}, X.resolution)$
13: **return** $\hat{Y}$

---

Larger frozen pre-trained encoders do not necessarily lead to better segmentation results. Therefore, the search space in Table 1 is proposed, based on the lightweight MobileNetV3 [9] architecture. The supernet pre-trained on ImageNet [8] from Once-for-All [5] is directly used for searching, without any additional training. Similarly to most convolutional architectures, the number of feature maps increases sequentially, while the resolution of the feature maps decreases sequentially (except for stage 3). For each stage, the expansion ratio and kernel size for the blocks in that stage are searched. Furthermore, the blocks to extract from and the kernel size (patch size) for average pooling of the extracted outputs are searched. It should be noted that the number of feature maps (network width) is additionally searched; however, this search is not performed separately for each stage because the supernet was not trained for that. Furthermore, the minimal depth of a stage is set to two blocks and its depth is only increased if a deeper block is extracted from the stage.

| Stage | Resolution | Width | Expansion Ratio | Kernel Size | Extraction Block | Patch Size |
|-------|-----------|-------|-----------------|-------------|------------------|------------|
| 0 | 1/4 | {24, 32} | {3, 4, 6} | {3, 5, 7} | {∅, 1, 2, 3, 4} | [1, 16] |
| 1 | 1/8 | {40, 48} | {3, 4, 6} | {3, 5, 7} | {∅, 5, 6, 7, 8} | [1, 16] |
| 2 | 1/16 | {80, 96] | {3, 4, 6] | {3, 5, 7} | {∅, 9, 10, 11, 12} | [1, 16] |
| 3 | 1/16 | {112, 136] | {3, 4, 6] | {3, 5, 7} | {∅, 13, 14, 15, 16} | [1, 16] |
| 4 | 1/32 | {160, 192] | {3, 4, 6] | {3, 5, 7} | {∅, 17, 18, 19, 20} | [1, 16] |

Table 1: The search space $\mathcal{A}$ based on the MobileNetV3 [9] architecture.

To automate the search process, it can be represented as a multi-objective black-box optimization problem:

$$\text{maximize} \quad PERFORMANCE(\boldsymbol{\theta}), \quad \text{minimize} \quad COMPLEXITY(\boldsymbol{\theta}), \quad \text{subject to} \quad \boldsymbol{\theta} \in \mathcal{A} \quad (5)$$

In an iterative manner, the black-box optimization algorithm proposes an architecture from the search space. The simple anomaly segmentation process first (i) populates the memory bank with patches from normal images in the training set and, subsequently, (ii) predicts anomaly scores for patches from normal and anomalous images in the validation set. With ground truth masks of anomalous regions and predicted scores, the performance is calculated and returned along with the complexity of the architecture as the objective values to the black-box optimization algorithm. Note that the memory bank is reset after each iteration. After performing this iterative search process for a specified number of trials, Pareto optimal trials are identified as those that cannot be improved in one objective without deteriorating the other. For an overview of the complete method, refer to Figure 1.

## 5 Experiments

### 5.1 Experimental Setup

The goal of this paper is to determine how closely aligned the Pareto optimal architectures discovered using a limited number of available anomalies are to the Pareto optimal architectures that would be discovered using a larger set of unseen anomalies. In other words, the goal is to evaluate the generalization of the search under limited anomalous examples. For each category in the original MVTec dataset [3], different validation sets are created depending on $k$, the number of available examples per type of anomaly. The original dataset is modified as shown in Table 2.

|  | Train | Test | Validation |
|---|---|---|---|
| Original | $n_{\text{train}}$ normal images | $n_{\text{test,normal}}$ normal images
$n_{\text{test},t}$ anomaly images of each type $t$ | |
| Modified | $n_{\text{train}} - n_{\text{test,normal}}$ normal images | $n_{\text{test,normal}}$ normal images
$n_{\text{test},t} - 4$ anomalous images of each type $t$ | $k{=}1$: $n_{\text{test,normal}}$ normal images
and the first anomaly image of each type
$k{=}2$: $n_{\text{test,normal}}$ normal images
and the first two anomaly images of each type
$k{=}4$: $n_{\text{test,normal}}$ normal images
and the first four anomaly images of each type |

Table 2: Modifcations to the MVTec [3] dataset to include validation sets with different numbers of available examples per type of anomaly.

Searches are performed for each validation set for each category with rwAP on the corresponding validation set as the objective. Additionally, searches are performed for each category with rwAP on the test set as the objective. The latter test set search serves as an oracle, to determine the Pareto optimal architectures that would be discovered using a larger set of unseen anomalies. MOTPE [11] is used as the black-box optimization algorithm. All searches are limited to 2000 trials and are repeated with $seed \in 0, 1, 2$. The average runtime is 7 hours and 39 minutes. The last 1000 trials result in only slightly better pareto-fronts. Running the required searches for the main experimental results costs roughly 1380 GPU hours. For hardware and implementation details, refer to Appendix A.

### 5.2 Unconstrained Computational Complexity

In Table 3, the results of the architectures with the highest performance are presented. Patch-Core [12] with the default WideResNet-50 [15] backbone is included as a baseline. The results of using the test set to search are included as an oracle. Note that for the validation set searches, the highest performance with respect to the validation set may not necessarily be the highest performance with respect to the reported performance on the unseen test set.

The highest performance architectures are by definition the highest complexity architectures on the Pareto front. The relationship between performance and complexity is non-linear, displaying

a long tail of diminishing returns on performance for higher complexities. The length of this tail varies between different seeded searches, resulting in widely varying computational complexities for the highest performance architectures. In practice, one would select an architecture on the Pareto front with the desired trade-off between performance and complexity. For the complete Pareto fronts, refer to Appendix B.

Performance can exhibit considerable variation between different seeds, given the presence of distinct optima in the search space that are specific to the anomalies in the validation set. Some of these optima might not translate well to the test set, while others might. Given that the search space cannot be completely explored in 2000 trials and the optimization algorithm is non-deterministic, the test set performance may significantly vary, even if different seeded searches yield similar validation set performance.

AutoPatch outperforms PatchCore [12] for 13/15 categories with $k=1$ example per type of anomaly, while the architectures discovered by AutoPatch require significantly lower computational complexity compared to the default WideResNet-50 [15] backbone used in PatchCore [12]. Although most categories benefit from more anomalies, others exhibit no improvement or even degradation in performance. Performance degradation may be attributed to the large variance in performance for different seeds. A selection of qualitative test examples with an example failure case where last-stage extraction leads to unexpected behavior for AutoPatch ($k=4$) in the *Carpet* category is highlighted in Appendix C.

| Category | WRN-50 PatchCore [12] | AutoPatch ($k=1$) | AutoPatch ($k=2$) | AutoPatch ($k=4$) | AutoPatch (*Test'*) |
|---|---|---|---|---|---|
| Carpet (5) | 59.7 (18.41) | 57.2±3.7 (0.42±0.03) | **65.0**±5.6 (0.37±0.05) | 55.1±17.5 (0.50±0.27) | 68.8±1.1 (0.34±0.06) |
| Grid (5) | 38.4 (18.41) | 56.2±9.4 (0.56±0.18) | 59.4±2.4 (0.29±0.15) | **66.5**±0.9 (0.35±0.19) | 69.8±4.4 (0.38±0.18) |
| Leather (5) | 51.3 (18.41) | 76.1±1.9 (0.30±0.07) | **77.0**±2.5 (0.38±0.04) | 73.7±2.9 (0.37±0.20) | 78.3±0.8 (0.24±0.05) |
| Tile (5) | 68.3 (18.41) | 81.8±1.6 (0.26±0.02) | 84.0±1.0 (0.19±0.04) | **84.8**±0.6 (0.21±0.10) | 85.5±1.0 (0.13±0.00) |
| Wood (5) | 63.1 (18.41) | **74.5**±4.7 (0.33±0.04) | 72.3±4.9 (0.34±0.12) | 72.5±5.1 (0.42±0.10) | 75.6±0.4 (0.27±0.19) |
| Bottle (3) | 81.1 (18.41) | 82.7±2.7 (0.39±0.25) | 85.9±0.8 (0.40±0.15) | **86.5**±0.7 (0.33±0.03) | 86.6±1.7 (0.43±0.17) |
| Cable (8) | 66.4 (18.41) | 68.4±3.1 (0.63±0.24) | 69.9±0.5 (0.38±0.07) | **71.1**±1.6 (0.35±0.14) | 73.1±0.6 (0.28±0.06) |
| Capsule (5) | 39.6 (18.41) | 73.2±0.1 (0.38±0.01) | 74.5±1.3 (0.41±0.08) | **74.7**±0.8 (0.54±0.24) | 74.0±1.0 (0.32±0.21) |
| Hazelnut (4) | 68.8 (18.41) | 79.2±2.8 (0.33±0.09) | **79.8**±1.5 (0.34±0.08) | 79.0±0.8 (0.39±0.01) | 83.9±0.7 (0.33±0.07) |
| Metal nut (4) | 89.1 (18.41) | 91.5±0.9 (0.45±0.10) | **92.3**±1.2 (0.77±0.32) | 92.1±0.6 (0.46±0.13) | 93.0±0.2 (0.41±0.07) |
| Pill (7) | 71.1 (18.41) | 77.1±0.4 (0.55±0.17) | 77.0±1.0 (0.56±0.20) | **78.9**±1.4 (0.50±0.33) | 81.3±0.4 (0.59±0.17) |
| Screw (5) | 35.9 (18.41) | 45.1±0.9 (0.30±0.11) | 55.2±5.4 (0.40±0.04) | **59.6**±2.6 (0.41±0.08) | 65.9±0.4 (0.59±0.30) |
| Toothbrush (1) | 34.8 (18.41) | 37.5±17.0 (0.18±0.05) | 49.5±5.5 (0.39±0.13) | **56.3**±3.6 (0.30±0.14) | 60.3±0.1 (0.31±0.16) |
| Transistor (4) | 69.7 (18.41) | 61.3±2.5 (0.34±0.10) | 71.0±1.7 (0.43±0.14) | **71.1**±4.5 (0.42±0.17) | 75.5±0.6 (0.55±0.23) |
| Zipper (7) | 64.6 (18.41) | 74.6±1.0 (0.60±0.20) | 77.2±1.4 (0.60±0.42) | 76.9±3.6 (0.54±0.15) | 77.5±0.9 (0.36±0.08) |
| Mean | 60.1 (18.41) | 69.1±3.5 (0.40±0.11) | 72.7±2.5 (0.42±0.14) | **73.3**±3.2 (0.41±0.15) | 76.6±1.0 (0.37±0.13) |

Table 3: The results of the architectures with the highest performance. All reported values are rwAP (%) on the test set and GFLOPS in parentheses. The category names are followed by the number of anomaly types for that category in parentheses.

### 5.3 Constrained Computational Complexity

In Table 4, the results for AutoPatch with a computational complexity constraint equal to the computational complexity of PatchCore [12] with a MobileNetV3-Large [9] backbone are presented. For fair comparison, the computationally expensive default WideResNet-50 [15] backbone for PatchCore [12] is replaced by the lightweight MobileNetV3-Large [9] backbone and extraction is performed similarly; from the last layer with 1/8 resolution (stage 1) and 1/16 resolution (stage 3).

Surprisingly, a backbone with 18x fewer FLOPS yields comparable mean performance for PatchCore [12]. Yet, AutoPatch outperforms PatchCore [12] in 11/15 categories with $k=1$ anomaly example, consistently maintaining lower computational complexity.

Notably, the performance of the constrained architectures doesn't significantly drop compared to the unconstrained ones in Section 5.2, confirming the relationship of diminishing returns on performance with increased complexity.

| Category | MNV3-L PatchCore [12] | AutoPatch (*k=1*) | AutoPatch (*k=2*) | AutoPatch (*k=4*) | AutoPatch (*Test'*) |
|---|---|---|---|---|---|
| Carpet (5) | 59.7 (0.31) | $58.8_{\pm6.4}$ ($0.27_{\pm0.02}$) | $\underline{64.7}_{\pm3.4}$ ($0.29_{\pm0.01}$) | $\mathbf{66.0}_{\pm1.2}$ ($0.27_{\pm0.03}$) | $68.6_{\pm1.5}$ ($0.30_{\pm0.01}$) |
| Grid (5) | 39.0 (0.31) | $58.5_{\pm7.9}$ ($0.22_{\pm0.02}$) | $\underline{59.8}_{\pm1.8}$ ($0.23_{\pm0.05}$) | $\mathbf{69.3}_{\pm2.8}$ ($0.23_{\pm0.07}$) | $68.8_{\pm4.0}$ ($0.27_{\pm0.04}$) |
| Leather (5) | 49.1 (0.31) | $\mathbf{76.2}_{\pm2.0}$ ($0.24_{\pm0.04}$) | $70.5_{\pm9.8}$ ($0.27_{\pm0.02}$) | $\underline{74.7}_{\pm3.6}$ ($0.22_{\pm0.08}$) | $78.3_{\pm0.8}$ ($0.24_{\pm0.05}$) |
| Tile (5) | 70.7 (0.31) | $81.8_{\pm1.6}$ ($0.26_{\pm0.02}$) | $\underline{84.0}_{\pm1.0}$ ($0.19_{\pm0.04}$) | $\mathbf{85.4}_{\pm0.6}$ ($0.15_{\pm0.00}$) | $85.5_{\pm1.0}$ ($0.13_{\pm0.00}$) |
| Wood (5) | 55.7 (0.31) | $\underline{72.2}_{\pm2.9}$ ($0.25_{\pm0.04}$) | $\mathbf{73.7}_{\pm2.9}$ ($0.23_{\pm0.08}$) | $72.1_{\pm3.1}$ ($0.25_{\pm0.05}$) | $75.6_{\pm0.4}$ ($0.21_{\pm0.09}$) |
| Bottle (3) | 83.8 (0.31) | $83.9_{\pm2.2}$ ($0.22_{\pm0.08}$) | $\mathbf{86.4}_{\pm0.1}$ ($0.23_{\pm0.01}$) | $\underline{86.3}_{\pm0.5}$ ($0.27_{\pm0.04}$) | $86.4_{\pm1.9}$ ($0.28_{\pm0.05}$) |
| Cable (8) | 69.0 (0.31) | $67.1_{\pm3.4}$ ($0.27_{\pm0.05}$) | $\underline{70.5}_{\pm0.5}$ ($0.29_{\pm0.03}$) | $\mathbf{71.1}_{\pm1.4}$ ($0.27_{\pm0.05}$) | $73.3_{\pm1.0}$ ($0.26_{\pm0.03}$) |
| Capsule (5) | 41.2 (0.31) | $72.8_{\pm0.9}$ ($0.27_{\pm0.01}$) | $\underline{73.3}_{\pm0.5}$ ($0.22_{\pm0.05}$) | $\mathbf{73.9}_{\pm0.1}$ ($0.28_{\pm0.03}$) | $73.8_{\pm0.7}$ ($0.23_{\pm0.07}$) |
| Hazelnut (4) | 71.9 (0.31) | $78.1_{\pm2.6}$ ($0.29_{\pm0.04}$) | $\mathbf{80.0}_{\pm3.0}$ ($0.26_{\pm0.01}$) | $\underline{79.1}_{\pm0.3}$ ($0.29_{\pm0.03}$) | $83.3_{\pm0.7}$ ($0.29_{\pm0.03}$) |
| Metal nut (4) | 90.9 (0.31) | $91.7_{\pm0.4}$ ($0.29_{\pm0.02}$) | $\underline{92.3}_{\pm0.6}$ ($0.25_{\pm0.05}$) | $\mathbf{92.5}_{\pm0.4}$ ($0.28_{\pm0.03}$) | $92.9_{\pm0.1}$ ($0.24_{\pm0.06}$) |
| Pill (7) | 66.7 (0.31) | $75.8_{\pm1.1}$ ($0.29_{\pm0.02}$) | $\underline{76.8}_{\pm0.4}$ ($0.29_{\pm0.02}$) | $\mathbf{78.1}_{\pm0.9}$ ($0.23_{\pm0.01}$) | $79.7_{\pm0.3}$ ($0.24_{\pm0.05}$) |
| Screw (5) | 39.5 (0.31) | $45.2_{\pm1.1}$ ($0.26_{\pm0.08}$) | $\underline{53.1}_{\pm8.3}$ ($0.21_{\pm0.04}$) | $\mathbf{60.3}_{\pm1.2}$ ($0.28_{\pm0.01}$) | $61.4_{\pm2.9}$ ($0.28_{\pm0.04}$) |
| Toothbrush (1) | 38.3 (0.31) | $37.5_{\pm17.0}$ ($0.18_{\pm0.05}$) | $\underline{51.5}_{\pm5.0}$ ($0.27_{\pm0.04}$) | $\mathbf{56.1}_{\pm3.6}$ ($0.25_{\pm0.06}$) | $60.2_{\pm0.2}$ ($0.17_{\pm0.05}$) |
| Transistor (4) | 64.0 (0.31) | $62.3_{\pm2.0}$ ($0.28_{\pm0.01}$) | $\mathbf{73.2}_{\pm2.3}$ ($0.27_{\pm0.03}$) | $\underline{71.1}_{\pm4.6}$ ($0.28_{\pm0.05}$) | $73.7_{\pm0.6}$ ($0.26_{\pm0.03}$) |
| Zipper (7) | 61.3 (0.31) | $72.4_{\pm3.1}$ ($0.29_{\pm0.02}$) | $\mathbf{75.4}_{\pm0.2}$ ($0.20_{\pm0.07}$) | $\underline{74.5}_{\pm3.8}$ ($0.24_{\pm0.04}$) | $77.0_{\pm0.8}$ ($0.28_{\pm0.02}$) |
| Mean | 60.1 (0.31) | $68.9_{\pm3.6}$ ($0.26_{\pm0.03}$) | $\underline{72.3}_{\pm2.7}$ ($0.25_{\pm0.04}$) | $\mathbf{74.0}_{\pm1.9}$ ($0.25_{\pm0.04}$) | $75.9_{\pm1.1}$ ($0.25_{\pm0.04}$) |

Table 4: The results of the discovered architectures with a computational complexity constraint of 0.31 GFLOPS. All reported values are rwAP (%) on the test set and GFLOPS in parentheses. The category names are followed by the number of anomaly types for that category in parentheses.

## 5.4 Metric Comparison

The Area Under the Receiver Operating Characteristic (AUROC) metric, while popular, has nuances that can complicate clear and accurate comparisons. Table 5 shows that AUROC, particularly without an upper limit on the False Positive Rate (FPR), tends to offer an overly optimistic evaluation of performance. Even with an FPR limit of 30%, as proposed by the authors of MVTec [3], AUROC may still overestimate performance.

In contrast, Average Precision (AP) is a more straightforward metric that does not necessitate adjustments in relation to FPR, thereby providing a more reliable performance indication. Furthermore, the region-weighted Average Precision (rwAP), which was the optimization metric of choice for this paper, ensures balanced evaluation. It assigns equal importance to regions of varying sizes and anomaly types, irrespective of their distinct region counts.

The results in Table 5 should be interpreted with the understanding that the optimization was based on rwAP. Therefore, AutoPatch (*k=1*) might not outperform PatchCore [12] when evaluated using the AUROC metric. Despite this, when evaluated using the widely recognized AP metric, AutoPatch shows superior performance over PatchCore [12] in all cases.

| Metric | WRN-50 PatchCore [12] | AutoPatch (*k=1*) | AutoPatch (*k=2*) | AutoPatch (*k=4*) | AutoPatch (*Test'*) |
|---|---|---|---|---|---|
| AUROC | 98.38 (3) | 98.26 (5) | 98.38 (2) | 98.32 (4) | 98.48 (1) |
| $AUROC_{0.3}$ | 96.95 (4) | 96.84 (5) | 97.09 (2) | 96.98 (3) | 97.30 (1) |
| AP | 63.98 (5) | 69.59 (4) | 71.99 (2) | 71.83 (3) | 75.09 (1) |
| rwAP | 60.13 (5) | 69.12 (4) | 72.69 (3) | 73.32 (2) | 76.66 (1) |

Table 5: Comparison of different metrics. The mean performance of all categories in the test set is evaluated as a percentage. The ranking with respect to the evaluated metric is given in parentheses.

## 5.5 Oracle Architectures

In Table 6, the highest performance architectures found with AutoPatch (*Test'*) are presented. The results indicate a wide array of optimal architectures across different categories, underscoring the imperative for category-specific neural architecture searches. Interestingly, a kernel size of 7 proves to be the most effective for all categories in the first stage when computational complexity isn't a limiting factor. However, in terms of other parameters and stages, few consistencies are observed.

| Category | Stages | | | | |
|---|---|---|---|---|---|
| | 0 | 1 | 2 | 3 | 4 |
| Carpet | B0;E4;K7;P5;W24 | B6;E3;K7;P1;W40 | B0;E3;K5;P7;W80 | B15;E3;K7;P9;W112 | |
| Grid | B3;E4;K7;P1;W32 | B7;E3;K7;P3;W48 | B12;E4;K7;P7;W96 | | |
| Leather | B2;E4;K7;P1;W24 | B6;E4;K7;P3;W40 | | | |
| Tile | B1;E3;K7;P1;W24 | B5;E3;K5;P4;W40 | | | |
| Wood | B0;E3;K7;P11;W24 | B5;E4;K5;P1;W40 | | | |
| Bottle | B1;E6;K7;P16;W24 | B5;E6;K7;P1;W40 | B10;E4;K7;P13;W80 | | |
| Cable | B4;E3;K7;P13;W24 | B5;E6;K7;P2;W40 | B10;E4;K7;P3;W80 | | |
| Capsule | B2;E6;K7;P13;W24 | B5;E4;K5;P1;W40 | | | |
| Hazelnut | B1;E6;K7;P1;W24 | B0;E3;K7;P5;W40 | B11;E3;K7;P10;W80 | B13;E3;K7;P3;W112 | |
| Metal Nut | B3;E6;K7;P2;W24 | B5;E6;K5;P3;W40 | B0;E3;K5;P8;W80 | | |
| Pill | B1;E6;K7;P3;W32 | B5;E4;K7;P1;W48 | B9;E3;K7;P6;W96 | | |
| Screw | B3;E3;K7;P2;W32 | B6;E4;K7;P1;W48 | B11;E6;K7;P7;W96 | B16;E6;K7;P16;W136 | B18;E3;K7;P13;W192 |
| Toothbrush | B2;E3;K7;P1;W24 | B5;E4;K5;P1;W40 | B0;E3;K7;P12;W80 | B0;E6;K5;P10;W112 | B20;E6;K7;P13;W160 |
| Transistor | B2;E6;K7;P4;W32 | B6;E3;K7;P3;W48 | B12;E3;K7;P3;W96 | B13;E3;K5;P13;W136 | |
| Zipper | B1;E6;K7;P1;W24 | B8;E6;K7;P3;W40 | | | |

Table 6: The highest performance oracle architectures found with AutoPatch (*Test'*) (*seed* = 1). *B*, *E*, *K*, *P*, *W* correspond to extraction block, expansion ratio, kernel size, patch size and width, respectively.

## 5.6 Search Space Ablation

In Table 7, the results are included for the highest performance architectures when the network width, stage kernel size, and stage expansion ratio are fixed to the maximum value in the search space (see Table 1). For both the full and fixed search space, the minimal depth of a stage is set to two blocks, and is increased only if a deeper block is extracted from the stage. Searches with *k=2* and *k=4* are skipped. The exact same experimental setup as described in Section 5.1 is followed.

The results show that searching for network width, stage kernel size, and stage expansion ratio results in significantly lower model complexity, but leads to a small degradation in mean performance. Performance degradation is attributed to the "harder" higher-dimensional optimization problem of the larger search space. The fixed search space does not lead to a better generalization under limited anomalous examples. This indicates that it is advisable to use the full search space, even under limited anomalous examples, if computational complexity is of concern.

## 6 Conclusion

This paper presented AutoPatch, a neural architecture search method designed to improve the efficiency of visual anomaly segmentation. With only one example per type of anomaly, the method successfully discovers architectures that not only exceed prior work in performance on the challenging MVTec [3] dataset, but also require fewer computational resources.

The research question central to this paper states: *"To what extent can the Pareto optimal neural architectures for visual anomaly segmentation be approximated, given limited anomalies?"*

| Category | AutoPatch ($k{=}1$) (full) | AutoPatch ($k{=}1$) (fixed) | AutoPatch (*Test'*) (full) | AutoPatch (*Test'*) (fixed) |
|---|---|---|---|---|
| Carpet (5) | $57.2_{\pm3.7}$ ($0.42_{\pm0.03}$) | $63.6_{\pm5.5}$ ($0.92_{\pm0.06}$) | $68.8_{\pm1.1}$ ($0.34_{\pm0.06}$) | $70.5_{\pm0.4}$ ($0.75_{\pm0.35}$) |
| Grid (5) | $56.2_{\pm9.4}$ ($0.56_{\pm0.18}$) | $58.6_{\pm8.2}$ ($0.88_{\pm0.32}$) | $69.8_{\pm4.4}$ ($0.38_{\pm0.18}$) | $72.7_{\pm0.1}$ ($0.66_{\pm0.25}$) |
| Leather (5) | $76.1_{\pm1.9}$ ($0.30_{\pm0.07}$) | $74.0_{\pm7.7}$ ($0.70_{\pm0.20}$) | $78.3_{\pm0.8}$ ($0.24_{\pm0.05}$) | $79.1_{\pm0.3}$ ($0.66_{\pm0.10}$) |
| Tile (5) | $81.8_{\pm1.6}$ ($0.26_{\pm0.02}$) | $79.4_{\pm0.7}$ ($0.51_{\pm0.02}$) | $85.5_{\pm1.0}$ ($0.13_{\pm0.00}$) | $83.0_{\pm0.0}$ ($0.42_{\pm0.00}$) |
| Wood (5) | $74.5_{\pm4.7}$ ($0.33_{\pm0.04}$) | $68.5_{\pm3.9}$ ($0.52_{\pm0.00}$) | $75.6_{\pm0.4}$ ($0.27_{\pm0.19}$) | $78.5_{\pm0.8}$ ($0.55_{\pm0.06}$) |
| Bottle (3) | $82.7_{\pm2.7}$ ($0.39_{\pm0.25}$) | $84.7_{\pm1.2}$ ($0.55_{\pm0.23}$) | $86.6_{\pm1.7}$ ($0.43_{\pm0.17}$) | $87.7_{\pm0.0}$ ($0.60_{\pm0.07}$) |
| Cable (8) | $68.4_{\pm3.1}$ ($0.63_{\pm0.24}$) | $67.5_{\pm1.4}$ ($1.17_{\pm0.13}$) | $73.1_{\pm0.6}$ ($0.28_{\pm0.06}$) | $71.7_{\pm0.3}$ ($0.76_{\pm0.33}$) |
| Capsule (5) | $73.2_{\pm0.1}$ ($0.38_{\pm0.01}$) | $74.3_{\pm0.5}$ ($0.61_{\pm0.05}$) | $74.0_{\pm1.0}$ ($0.32_{\pm0.21}$) | $75.0_{\pm0.0}$ ($0.98_{\pm0.31}$) |
| Hazelnut (4) | $79.2_{\pm2.8}$ ($0.33_{\pm0.09}$) | $80.2_{\pm1.3}$ ($0.96_{\pm0.16}$) | $83.9_{\pm0.7}$ ($0.33_{\pm0.07}$) | $84.2_{\pm0.3}$ ($0.81_{\pm0.15}$) |
| Metal nut (4) | $91.5_{\pm0.9}$ ($0.45_{\pm0.10}$) | $91.4_{\pm0.8}$ ($0.98_{\pm0.07}$) | $93.0_{\pm0.2}$ ($0.41_{\pm0.07}$) | $94.1_{\pm0.2}$ ($0.73_{\pm0.18}$) |
| Pill (7) | $77.1_{\pm0.4}$ ($0.55_{\pm0.17}$) | $77.2_{\pm0.7}$ ($0.86_{\pm0.13}$) | $81.3_{\pm0.4}$ ($0.59_{\pm0.17}$) | $81.9_{\pm0.0}$ ($0.61_{\pm0.08}$) |
| Screw (5) | $45.1_{\pm0.9}$ ($0.30_{\pm0.11}$) | $50.7_{\pm12.2}$ ($0.85_{\pm0.05}$) | $65.9_{\pm0.4}$ ($0.59_{\pm0.30}$) | $65.7_{\pm0.4}$ ($1.01_{\pm0.14}$) |
| Toothbrush (1) | $37.5_{\pm17.0}$ ($0.18_{\pm0.05}$) | $49.8_{\pm0.7}$ ($0.88_{\pm0.08}$) | $60.3_{\pm0.1}$ ($0.31_{\pm0.16}$) | $59.0_{\pm0.1}$ ($1.03_{\pm0.15}$) |
| Transistor (4) | $61.3_{\pm2.5}$ ($0.34_{\pm0.10}$) | $58.7_{\pm2.2}$ ($1.09_{\pm0.08}$) | $75.5_{\pm0.6}$ ($0.55_{\pm0.23}$) | $73.9_{\pm0.1}$ ($1.02_{\pm0.34}$) |
| Zipper (7) | $74.6_{\pm1.0}$ ($0.60_{\pm0.20}$) | $76.9_{\pm1.8}$ ($0.92_{\pm0.12}$) | $77.5_{\pm0.9}$ ($0.36_{\pm0.08}$) | $80.4_{\pm0.1}$ ($0.84_{\pm0.12}$) |
| Mean | $69.1_{\pm3.5}$ ($0.40_{\pm0.11}$) | $70.4_{\pm3.3}$ ($0.83_{\pm0.11}$) | $76.6_{\pm1.0}$ ($0.37_{\pm0.13}$) | $77.2_{\pm0.2}$ ($0.76_{\pm0.18}$) |

Table 7: Ablation study on the effect of using the full search space versus a search space with fixed width, kernel size, and expansion ratio. All reported values are rwAP (%) on the test set and GFLOPS in parentheses. The category names are followed by the number of anomaly types for that category in parentheses.

With one or two examples per type of anomaly, most categories show a relatively close approximation of the Pareto optimal architectures that would be discovered using a larger set of unseen anomalies.

However, reliable neural architecture search for visual anomaly segmentation remains a complex challenge. Although the application of neural architecture search with limited anomalous examples has shown promise, its effectiveness depends on the specific anomalies available and can exhibit considerable variance in real-world performance.

## 7 Limitations and Future Work

Reducing the memory bank to a coreset may further reduce overall computational complexity [12]. However, performing this process for each architecture in the search space comes at the expense of a slower search process. Therefore, AutoPatch uses full memory banks for searching and reducing the memory bank to a coreset may be performed after the search using the found optimal architecture. Searching the optimal subsampling ratio, while directly optimizing for inference time, as opposed to computational complexity, is an interesting avenue for future work.

Another practical consideration is the selection of an appropriate threshold for anomaly segmentation based on anomaly scores. Although this paper does not delve into this topic, future work could explore threshold selection in a similar manner, by using a representative validation set of anomalies.

Synthesizing and augmenting anomalies could offer a promising direction for future research, potentially leading to the development of more representative validation sets that better reflect the true distribution of anomalies and, subsequently, resulting in the discovery of more reliable architectures with lower performance variance.

Finally, this paper focused on examining the relationship between the number of anomalies used during the search process and the generalizability of the discovered architectures when applied to unseen anomalies. As a result, extensive ablation studies were not conducted in various search spaces and black-box optimization algorithms, leaving room for further exploration in these areas.

## 8 Broader Impact

The results presented in this paper highlight the potential of automated machine learning to optimize throughput in industrial quality control. Using automated machine learning techniques, AutoPatch demonstrates the ability to efficiently segment visual anomalies, which can be beneficial for various industries in detecting and addressing manufacturing defects or irregularities. This could lead to improved product quality, increased safety, and reduced costs associated with recalls or waste. Furthermore, AutoPatch does not rely on training, which saves the computational costs and resources typically associated with training a supernet, making it more accessible and environmentally friendly. After careful reflection, the authors have determined that this paper presents no notable negative impacts to society or the environment.

**Acknowledgements**. Sincere thanks to ASMPT Ltd for funding and facilitating this work.

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

## A  Hardware and Implementation Details

The MobileNetV3 [9] supernets with 1.0x and 1.2x width pre-trained on ImageNet [8] from Once-for-All [5] are used. Instead of cropping as in PatchCore [12], images are only resized to 224x224, as some anomaly regions may be (partially) cropped out otherwise.

For the searches, the multivariate MOTPE algorithm [4, 11] as implemented in Optuna [1] is utilized. The memory bank is not subsampled as subsampling for each trial would be considerably slow (subsampling may be performed after searching). All searches are performed with an NVIDIA GeForce RTX 2080Ti GPU. After loading the initial data onto the GPU, the CPU is no longer used for tensor operations.

For the PatchCore [12] baseline, the original author's code is used. The train and test set are modified to correspond to those described in 5.1. Cropping is disabled, and only resizing is performed. Instead of only measuring pixel-wise AUROC without any FPR limit, the code is adjusted to additionally measure AUROC limited to an FPR of 0.3, AP and rwAP.

## B  Full Pareto Fronts

In Figure 2, the full Pareto fronts are plotted for separate searches on the corresponding validation and test sets. The Pareto front architectures for the validation and test search are defined as those that have Pareto optimal complexity and performance on the validation and test sets, respectively. The y-axis represents rwAP (%) on the test set and the x-axis represents GFLOPS. As averaging Pareto fronts over different seeds is non-trivial, only the results for *seed=1* are shown. Unlike the main results in Section 5.2, this section includes more test images depending on $k$. $Test'$ thus consists of $n_{\text{test,normal}}$ normal images and $n_{\text{test},t} - k$ anomaly images of each type $t$.

For this specific seed, for 9/15 categories the Pareto optimal architectures with respect to the test set are well approximated by searching on the validation set with $k=1$ per type of anomaly. For 4/15 categories, the validation Pareto front moves closer to the test Pareto front by increasing $k$, indicating that more anomalous examples are necessary. For *zipper*, only $k=2$ leads to a good approximation, but in other seeds both $k=2$ and $k=4$ lead to a good approximation. For carpet, no $k$ leads to a good approximation, but in other seeds $k=2$ and $k=4$ do lead to a good approximation. The reasoning for this variance is given in Section 5.2.

## C  Qualitative Examples

Figure 3 presents a selection of qualitative test examples that compare PatchCore [12] and AutoPatch. The segmentation results of the best-performing architecture, similarly to Section 5.2, are shown for AutoPatch. The first two rows include examples where AutoPatch ($k=1$) demonstrates more fine-grained segmentations of anomalous regions compared to PatchCore [12]. The third row presents an example where AutoPatch ($k=1$) detects a scratch on a capsule, while PatchCore [12] fails to do so.

The final row shows a failure case for AutoPatch ($k=4$), where the entire test image is assigned a high anomaly score. The resulting architecture from only one of the three different seeded searches shows this phenomenon in two specific test images, but none of the validation images. This architecture stands out because it extracts from the final stage of the network (stage 4), whereas the other architectures for the carpet category do not. The final stage extraction resulted in slightly better segmentation results in the validation set. However, it is hypothesized that this particular hole in the carpet category test set resembles a class in ImageNet, causing the image to trigger high activation for more ImageNet-biased features in the final stage of the network.

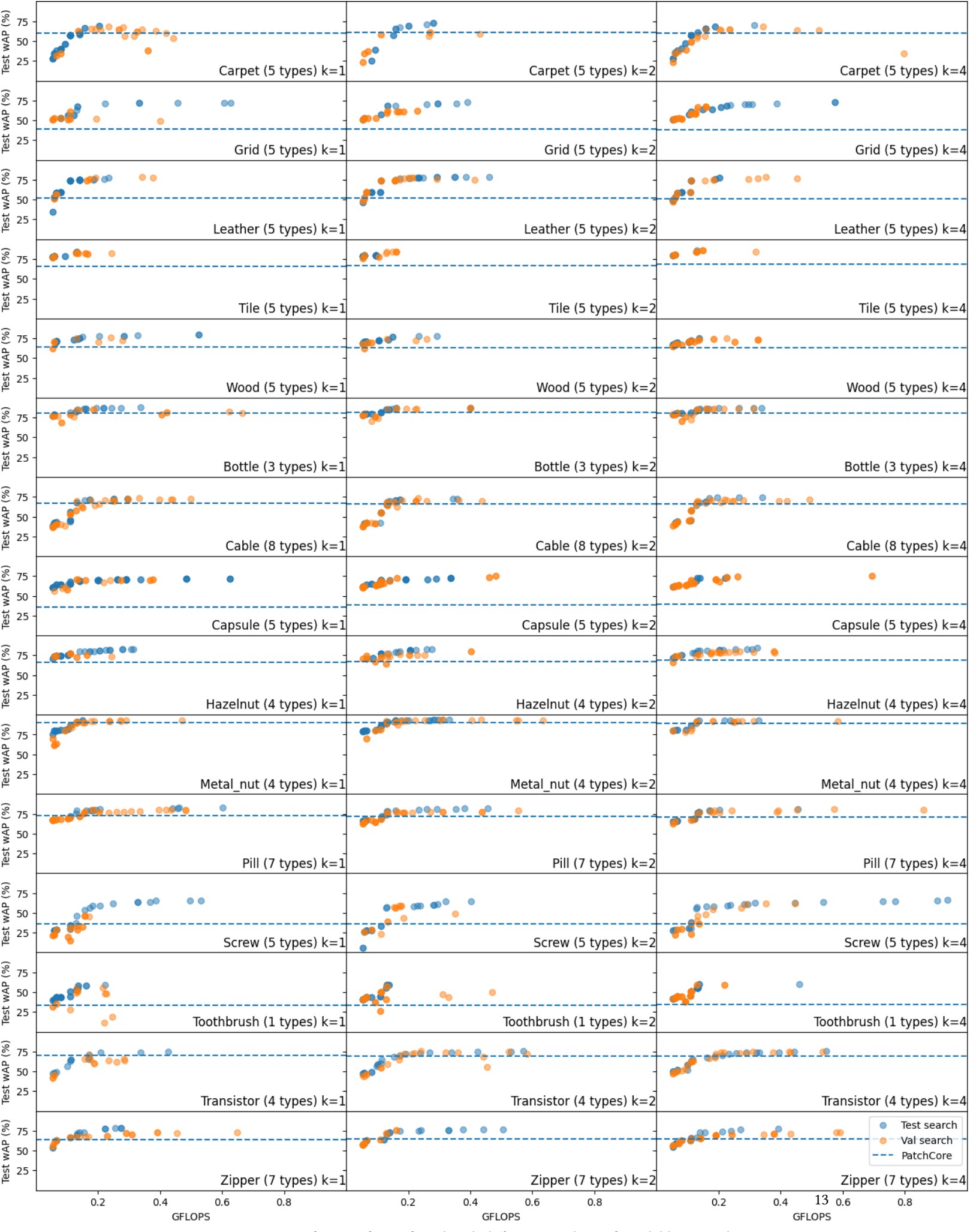

Figure 2: Comparison of Pareto fronts found with different numbers of available anomalies per type.

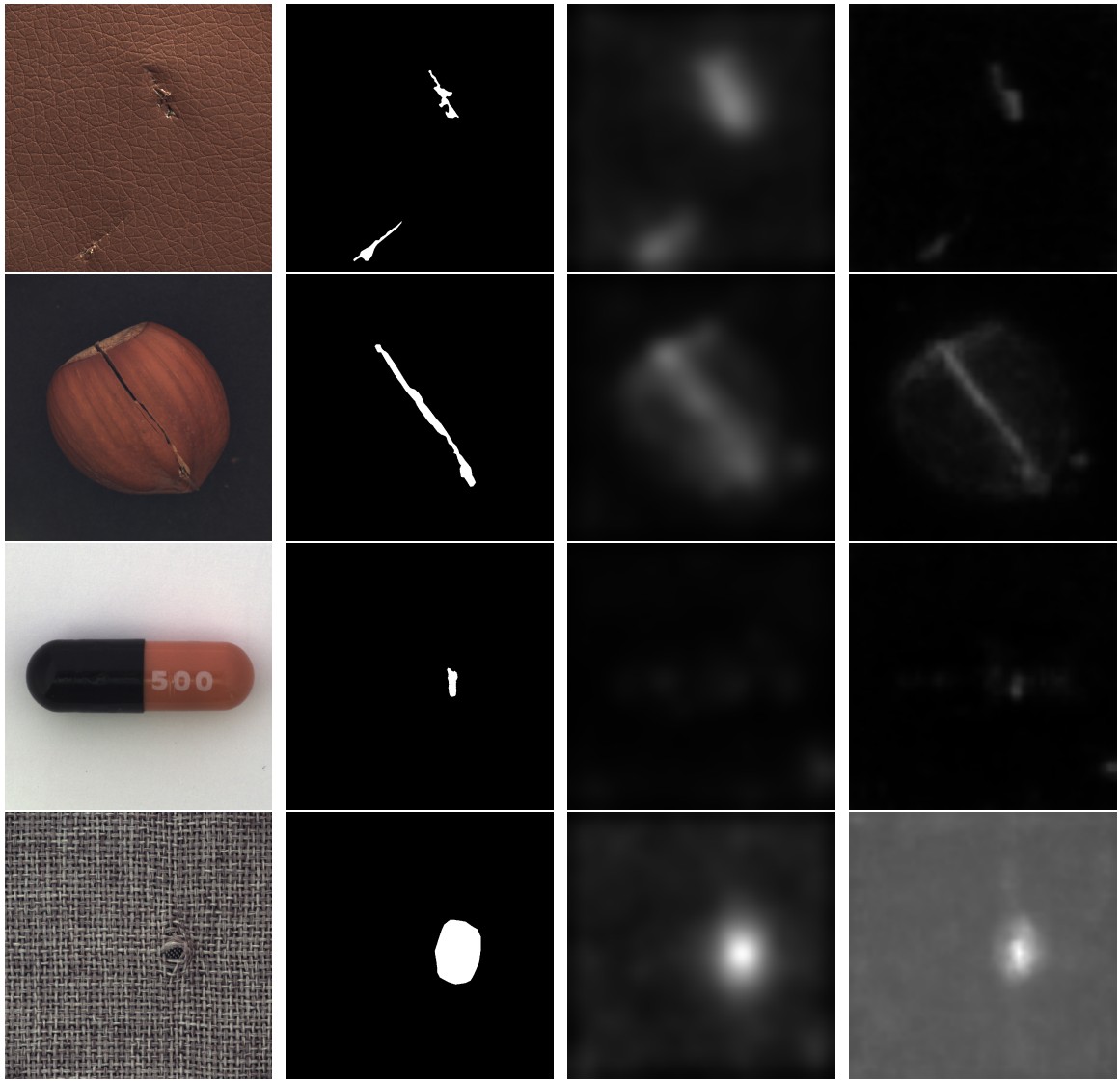

Figure 3: Qualitative examples comparing PatchCore [12] (3rd column) and AutoPatch (4th column). The anomaly scores are scaled by the minimum and maximum anomaly scores in the test set, where the minimum is a fully black pixel and the maximum is a fully white pixel.

