# OpenReview forum: "Neural Architecture Search for Visual Anomaly Segmentation"
_automl.cc/AutoML/2023/Conference — AutoML 2023 MainTrack_

### Official Review · Reviewer_63A5 · 2023-04-12

**Potential Impact On The Field Of Automl Rating:** 2
**Technical Quality And Correctness Rating:** 2
**Clarity Rating:** 3

**Summary Of Contributions:**

In this paper, the authors propose an architecture search method, AutoPatch, which is designed to search for effective segmentation algorithms for anomalies, especially under the scenario of only having relatively few anomalies. With this, they begin by proposing a new weighted average precision (wAP) metric to serve as an alternative to AUROC and AUPRO. The motivation for this is in some industrial applications you might not be able to set some fixed False Positive Rate (FPR) and wish to be flexible over multiple thresholds. The authors then propose an extension of Once-for-All to their scenario where they can extract features from training images, without any additional training, to help segment images during inference, and use MOTPE to maximize their proposed weighted mAP metric subject to minimize the number of flops used. The authors then analyze their method in comparison with an existing method for the problem of anomaly segmentation across different amounts of anomalies.

**Actions Required To Increase Overall Recommendation:**

I am happy to work with the authors to consider increasing my score if the authors can help me better assess the quality of this paper:

1. How does their method compare to their baseline using the metrics that were proposed by PatchCore?

2. How many frames are really necessary to store in order to attain good performance? This seems to be a fundamental question that I didn't see answered in the main text.

3. Does their method suffer from issues in generalization under even small amounts of domain shift?

4. I'm curious as to the effect of the selection of examples chosen for k = 1, 2, 4. I understand that this randomization is done over seeds but am wondering if there was additional work to verify the average performance over different k for both the proposed model and the baseline. Additionally, were the same examples used between the proposed method and the baseline in the evaluation?

**Clarity:**

I believe that the paper is for the most part presented in a very clear way; however, as alluded to above, the choice of 'wAP' for the name of the proposed metric is confusing (unless I'm mistaken and this refers to what other people would call wAP) because other people might be thinking of another thing. Additionally, I think the presentation of the analysis of computational efficiency is confusing, as they seem to all be crammed into a very large figure on page 9 that is difficult to parse.

**Overall Review:**

Pros:
+ The wAP metric seems well-motivated and is an unintuitive way to tackle the problem of class imbalance under the scenario when you don't have a fixed false positive rate.
+ The neural architecture search seems very thorough and the formulation of solving their black-box optimization scheme with MOTPE.
+ The author's proposed method doesn't require any additional training, something that I believe is underplayed
+ The authors do a good job of answering their proposed question of "How close can they get to the Pareto frontier'.

Cons:
- The combination of the new wAP metric alongside proposing the neural architecture search scheme slightly complicates the comparison to the baseline chosen.
- The use of the name 'Weighted Average Precision' seems overloaded to something that is common elsewhere in the literature and refers to something different which is slightly confusing.
- One of the motivations in section 3 is "However, these metrics lack flexibility for diverse requirements and could potentially lead to inaccurate performance indications in the absence of an appropriate FPR limit"; however, it's unclear from the setup that there exist such scenarios where an ~some FPR limit cannot be set that is semi-optimal.
- The use of the memory bank here seems to be a major limitation because the amount of memory required to both store the examples and the amount of computing required to search them is linear in the amount of training data. The authors do recognize this in their limitations; however, I could not find details of the necessary size of the memory bank that was used to achieve their results and I think that might be a crucial factor in their performance.
- Unless I misunderstood, it seems like this method is very prone to failing under domain shift, where images are at inference.


**Potential Impact On The Field Of Automl:**

Since this paper is targeted at a very particular problem within AutoML, namely doing a neural architecture search for anomaly segmentation, I don't think that it stands to make a very large impact in the field of AutoML because it fits a particular niche and the goal is to beat the state of the art in that problem. Additionally, I think wAP could maybe be an interesting contribution in the scope that it's agnostic to all problems which don't have a fixed FPR; however, the fact that it overlead the naming from a metric that already exists means that it will probably not be recognized.

**Review Confidence:**

4: You are confident in your assessment, but not absolutely certain. It is unlikely, but not impossible, that you did not understand some parts of the submission or that you are unfamiliar with some pieces of related work.

**Review Rating:**

6: Borderline Leaning Accept: Technically sound paper where reasons to accept outweigh reasons to reject. Please use sparingly.

**Review Summary:**

Overall, my current recommendation for this paper is a Weak Reject. The authors in this work solve a specific problem with pretty intuitive architectural search solutions and introduce a nice metric to help address their setup of anomaly detection. The main weaknesses of the paper stem from the lack of comparison using traditional metrics to their chosen baseline and some issues with their modeling. While they don't need to include any additional training, they have a method that is bandlimited by the number of training frames you can store, which makes it also seemingly brittle to domain shift. I also strongly believe that wAP here is causing some confusion and that the authors could benefit by changing the naming. I could be misunderstanding something here about their intended use case, and do think that for their problem setup, the authors offer a strong piece of work.

**Technical Quality And Correctness:**

I believe that the level of architecture search as well as the method for optimizing it both are of high quality. I appreciate the fact that the authors pay a great amount of detail to the train/val/test split for their optimizations and evaluation procedures. One major technical issue with this work is that the authors propose this new optimization scheme alongside their new metric wAP, and then go on to evaluate their new method on the new metric. This raises the question, is the method just good for this particular metric? Do the results change when different metrics are chosen? In comparison to PatchCore, it seems unfair to not use the same evaluation criteria that PatchCore used. Finally, because the strategy at inference is to compare to training patches to do anomaly detection, isn't it true that this would suffer massively from domain shift (even small ones), where everything would be classified as an anomaly?

---

> ### Author Response · Authors · 2023-05-01
> **Response to Reviewer 63A5**
>
> Dear Reviewer 63A5,
>
> Thank you for your review of the paper. We appreciate your feedback and have addressed your concerns in the revised version of the paper.
>
> **Metric Naming:** We acknowledge the confusion caused by the name "weighted Average Precision (wAP)" and have changed it to "region-weighted Average Precision (rwAP)" to prevent confusion and emphasize that weighting is done over regions. This name is generic, as the metric may indeed be useful outside of visual anomaly segmentation.
>
> **Fair Evaluation:** In the initial version of the paper, all experiments were performed using the newly introduced region-weighted Average Precision. However, we understand your concern regarding the potential disadvantage of the compared baseline due to the new metric. Thus, we have moved the comparison table using existing metrics, including AUROC without an FPR limit as used by PatchCore, from the Appendix to Section 5.4 in the revised version, along with a more elaborate analysis. Additionally, this section highlights a scenario where a semi-optimal FPR limit makes evaluation troublesome.
>
> **Domain Shift:** The robustness to domain shifts of AutoPatch should be comparable to PatchCore, and it is not the main focus of this work. However, it is hypothesized that ImageNet features do show some robustness to domain shift. We refer to the following work that finds PatchCore segmentation performance is competitive under domain shifts: **[https://arxiv.org/pdf/2304.02216.pdf](https://arxiv.org/pdf/2304.02216.pdf)**
>
> **Efficiency Analysis:** We agree that the large figure with the Pareto fronts was crowded and difficult to interpret. Therefore, we have moved this figure to the Appendix and added Section 5.3 to evaluate the performance of architectures under a computational complexity constraint. This section includes an easier-to-parse table with an additional comparison to PatchCore with a MobileNetV3-Large backbone.
>
> **Memory Bank:** Your point about the memory bank is valid, and we acknowledge that the memory bank size is crucial to the amount of compute required. Subsampling as introduced by PatchCore allows to maintain a sufficient memory size to query from, instead of the whole training set, which is intractable for larger real-world datasets of normal examples. The results presented in this paper are without subsampling, both for PatchCore and AutoPatch. As a result, the size of the memory bank is equal to the number of patches in the training set for each category (see Section 5.1 Experimental Setup). We have included a better elaboration on the memory bank and subsampling in Section 7 Limitations and Future Work of the revised version. AutoPatch allows subsampling of the memory bank with the found architecture after completing the search. Performance is not significantly affected up to a certain subsampling ratio, as observed by PatchCore. To answer your question on how many frames are needed to attain good segmentation performance, it is between 1% to 10% of all patches from the training sets in MVTec (Figure S6 in the supplementary of PatchCore).
>
> **Validation Sets:** The images included in the validation set are not based on the seed, as they are not random. As described in Section 5.1 Experimental Setup, the validation sets are the first 1, 2, or 4 images of the corresponding anomaly type of the corresponding category in MVTec. PatchCore uses the full test sets of anomalies for all categories to determine a general choice of extraction layers and patch size. However, access to a large test set of anomalies is unrealistic. Therefore, the AutoPatch paper proposes to use small category-specific validation sets instead and shows that the optimal extraction layers and patch sizes are data-dependent and that one configuration, as proposed by PatchCore, does not rule all categories.
>
> We hope these revisions address your concerns and improve the quality of our paper.
>
> Thank you for your valuable feedback.

---

> > ### Comment · Reviewer_63A5 · 2023-05-01
> > **Thank you for your considerate response!**
> >
> > I appreciate your careful consideration of my comments.
> >
> > I believe that with the changes that you have proposed that this work now merits acceptance to the conference, but the fact that the memory bank is a significant limitation of the method, that additional results with the subsampling, I believe that simply the comparison and improvement over the current state of the art with PathCore merits a minor accept. Another thing that influences this decision is that while the goal of patch core isn't necessarily domain shift, the intended industrial use case would suggest that it would be necessarily robust (similar to the citation of PatchCore). I think this would be a valuable future contribution.
> >
> > Validation Set Response
> > This didn't play a significant role in my final decision, but I am slightly confused as to your clarification about validation sets. In comparison to PatchCore, wouldn't it be a more fair comparison to evaluate both methods against the same sets of examples? Perhaps I am misunderstanding something about your evaluation.

---

> > > ### Author Response · Authors · 2023-05-02
> > > **Response to Reviewer 63A5**
> > >
> > > Dear Reviewer 63A5,
> > >
> > > Thank you for your updated assessment of our paper. We are glad to see that our revisions have addressed some of your concerns.
> > >
> > > Regarding the memory bank, we acknowledge that this is a crucial aspect of our method, and we will consider conducting more experiments with subsampling in the future to further analyze its impact on performance.
> > >
> > > As for the validation set question, our aim in using category-specific validation sets is to demonstrate that one configuration, as proposed by PatchCore, does not rule all categories. AutoPatch is designed to adapt to different categories and find optimal extraction layers and patch sizes specific to the validation data it encounters. PatchCore's method assumes a general predetermined choice for extraction layers and a single patch size. Both methods are trained on the same train set and evaluated on the same test set. However, it's non-trivial how to use the validation images for PatchCore.

---

### Official Review · Reviewer_LNNk · 2023-04-12

**Potential Impact On The Field Of Automl Rating:** 3
**Technical Quality And Correctness Rating:** 3
**Clarity:** The manuscript is well-written and ea…
**Clarity Rating:** 3

**Summary Of Contributions:**

In this paper, authors applied neural architecture search to the visual anomaly segmentation problem to optimize the neural architecture for the best weighted Average Precision (wAP) performance with respect to FLOPS. The proposed AutoPatch utilizes MobileNetV3 as its backbone and optimizes various width, expansion ratio, kernel size, extraction block, and patch size values for each stage using FLOPS and wAP based Multi-Objective Tree-structured Parzen Estimator (MOTPE). The experimental results show that AutoPatch outperforms the existing state-of-the-art (SOTA) PatchCore on the MVTec dataset, demonstrating superior performance.

**Actions Required To Increase Overall Recommendation:**

I think it would be recommended to include comparisons with techniques other than PatchCore, comparison experiments on different datasets, and a thorough ablation study.

**Overall Review:**

The strengths of this paper lie in finding a highly efficient neural network structure for visual anomaly segmentation using neural architecture search, as well as proposing a new metric for performance measurement and increasing the learning efficiency to enable learning with fewer anomaly samples. The proposed technique has demonstrated performance surpassing the existing state-of-the-art, PatchCore, in experimental results. However, the lack of evaluation on multiple datasets and the absence of ablation study act as limitations.

**Potential Impact On The Field Of Automl:**

This paper holds significant implications for the application of AutoML techniques or NAS to various deep learning problems.

**Review Confidence:**

3: You are fairly confident in your assessment. It is possible that you did not understand some parts of the submission or that you are unfamiliar with some pieces of related work.

**Review Rating:**

7: Weak Accept: Technically sound paper with moderate-to-high impact and strong evaluation, with perhaps some minor flaws.

**Review Summary:**

Although this paper has some shortcomings in experimental design and performance validation, it is technically correct, and the proposed technique consistently demonstrates better performance compared to the comparing methods. Therefore, my recommendation is a weak accept.

**Technical Quality And Correctness:**

The authors suggest a suitable search space and conduct multi-objective NAS correctly for applying neural architecture search to the visual anomaly segmentation problem. However, it is regrettable that the experiments and results lack comparisons with other datasets that were included in the PatchCore paper, as well as insufficient ablation studies.

---

> ### Author Response · Authors · 2023-05-01
> **Response to Reviewer LNNk**
>
> Dear Reviewer LNNk,
>
> We appreciate your time and effort in reviewing our paper and providing valuable feedback.
>
> **Regarding Other Datasets:** We agree that diversifying the datasets used for testing would be beneficial. It's worth noting that we have treated MVTec as a meta-dataset, running separate experiments for each of its 15 distinct categories. This approach effectively means we've tested our method across 15 different datasets. We understand, however, the value of further diversifying the datasets used and will reserve this for our future work.
>
> **About Ablation Studies:** We took your feedback into account and have included a more thorough ablation of the method in the revised version of the paper. We have moved the ablation study on the search space from the Appendix to Section 5.6 and have included a comparison with PatchCore using a MobileNetV3-Large backbone in Section 5.3. We believe these additions will provide a more comprehensive understanding of our method's effectiveness.
>
> **Concerning Other Techniques:** To the best of our knowledge, our work is the first to directly apply neural architecture search to the task of visual anomaly segmentation/detection. Therefore, there isn't any other method that we could compare ours to in this context. We believe that our approach has opened up a new area of research in this field and look forward to more works in this direction that we can compare with in the future.
>
> We hope these responses address your concerns and further clarify the contributions of our paper.
>
> Once again, we thank you for your insightful feedback.

---

### Official Review · Reviewer_nyA3 · 2023-04-13

**Potential Impact On The Field Of Automl Rating:** 2
**Technical Quality And Correctness Rating:** 3
**Clarity Rating:** 3

**Summary Of Contributions:**

The authors propose AutoPatch, a NAS method that efficiently segments visual anomalies. It proposes a new metric, weighted average precision (wAP), to measure anomaly segmentation quality, and shows that AutoPatch outperforms the current state-of-the-art method with significantly fewer computational resources using only one example per anomaly type. The research question is how well Pareto optimal neural architectures can be approximated with limited anomalies. The authors conclude that while promising, reliable NAS for anomaly segmentation remains a complex challenge that depends on the specific anomalies available and can exhibit considerable variance.

**Actions Required To Increase Overall Recommendation:**

The authors could improve the paper by giving more insights into the search method used in the paper by comparing it to other search methods.

The authors should also give more details on the efficiency of the search algorithm. What is the search time?

What is the reason for AutoPatch to obtain better performance than PatchCore? Is it because of the search space or the search method?

The authors talk about inference time multiple times in the paper. However, there is no mention of the inference time of the searched models. The inference time should also be compared with PatchCore and previously searched or manually designed models.

It is also highly recommended to show the architecture of searched models for different scenarios. The authors did not discuss how the searched architectures look or explain them.

**Clarity:**

This paper is a straightforward application of existing multi-objective optimization methods on anomaly segmentation. The contributions with respect to the search algorithm are highly limited. The search space (Mobilenetv3) is directly inherited from the previous works, which is fine, but what are the main contributions in terms of the search algorithm and evaluation method? This paper could be well-suited for other ML venues as it has good contributions outside NAS.

Line 125: “Once-for-All network [5] being the first notable example.” This is arguable because there exists many weight-sharing works before OFA.

References are not needed in the abstract. The abstract should be standalone and must be able to be read without touching any other paper.

Although Section 4 talks about the proposed weighted average precision, it does not directly benefit the search method. It is just an evaluation metric developed to support anomaly segmentation.



**Overall Review:**

Positives: The application of NAS on Anomaly Segmentation is interesting. The presented approach is technically sound.
The empirical results presented in this paper are attractive and prove that NAS methods on Anomaly Segmentation are effective.

Negatives: The authors compared their work only with one previous method. Is it the only method in the literature? It is also unclear how to construct a small size of the training set with one example per anomaly type.

**Potential Impact On The Field Of Automl:**

The contributions of this paper in terms of NAS and AutoML are limited. However, the applied ML research community can find it interesting. It can have a great impact on anomaly segmentation problems, which itself a problem of importance.

**Review Confidence:**

3: You are fairly confident in your assessment. It is possible that you did not understand some parts of the submission or that you are unfamiliar with some pieces of related work.

**Review Rating:**

7: Weak Accept: Technically sound paper with moderate-to-high impact and strong evaluation, with perhaps some minor flaws.

**Review Summary:**

The reviewer is confident of the review. The reviewer has reviewed the search space, algorithm, and evaluation criteria to check the correctness. The reviewer has concerns with the novelty of the approach taken to search the neural architecture.

**Technical Quality And Correctness:**

The paper is well-written and easy to understand the main contributions. The results show an improvement over the baseline PatchCore. The search method and proposed metric sound correct. The exhaustive experiments and summarization of results look promising.

---

> ### Author Response · Authors · 2023-05-01
> **Response to Reviewer nyA3**
>
> Dear Reviewer nyA3,
>
> We greatly appreciate your thoughtful review of our paper and the suggestions you've provided to improve the quality of our work.
>
> **Regarding the Contributions:** We concur that our search space builds upon existing works such as Once-for-All and MobileNetV3. However, our approach introduces novel aspects, such as searching for each stage, extraction block, and patch size. This allows for a more dynamic and adaptable methodology than previous methods, like PatchCore, which uses a single patch size and a complicated pooling strategy. We also innovatively utilize dynamic stage depth to further reduce computational complexity. We have attempted to clarify these points in our manuscript to highlight the originality of our approach.
>
> **In relation to Once-for-All:** We agree with your point, and the statement in the paper has been amended to avoid any possible misunderstanding.
>
> **About the Abstract:** Your feedback on the abstract is valuable, and we have made the necessary revisions to ensure it stands alone without needing specific knowledge of other works.
>
> **On the Method:** We agree that our method can work with other performance metrics and black-box optimization algorithms, and we have updated the method section to reflect this.
>
> **In terms of Previous Methods:** To the best of our knowledge, our work is the first to directly apply neural architecture search to the task of visual anomaly segmentation/detection. Therefore, there isn't any other method that we could compare ours to in this context. We believe that our approach has opened up a new area of research in this field and look forward to more works in this direction that we can compare with in the future.
>
> **About the Validation Set:** We apologize for any confusion, and have clarified in Section 5.1 of our revised manuscript that the training set consists solely of normal images. The validation set is constructed by taking one anomalous example of each type of anomaly from the MVTec dataset. For example, for the bottle category, the k=1 validation set consists of 3 anomalous images (one broken_large, one broken_small, one contamination example) along with 19 good images. We believe that the updated table in Section 5.1 should offer a clearer understanding of how we split the data.
>
> **Considering the Novelty:** The novelty resides in the application of NAS to the visual anomaly segmentation problem. Specifically, we have innovated by using a pre-trained supernet, substantially reducing the computational load usually associated with NAS. This paper presents the first application of NAS to this demanding task. We also emphasize the value of our method's simplicity, as it can be effortlessly adapted and applied to real-world datasets.
>
> **Regarding the Search Time:** We have included the exact average search time in Section 5.1 of our revised manuscript.
>
> **On Better Performance:** We have included additional experiments to investigate why AutoPatch outperforms PatchCore:
>
> - Comparison with PatchCore with a MobileNetV3-Large backbone (Section 5.3)
> - An ablation study on the search space (moved from Appendix to Section 5.6)
>
> **In terms of Inference Time:** We agree with your observation. Given that the focus of this paper is computational complexity and not inference time, we have revised the manuscript to remove mentions of inference time.
>
> **About Found Architectures:** We have added a new section (Section 5.5) in our revised manuscript to describe the exact configurations of the architectures found for each category, providing insights into the oracle architectures.
>
> We hope that these revisions have addressed your concerns and further improved the quality of our paper.
>
> Once again, thank you for your valuable insights.

---

> > ### Comment · Reviewer_nyA3 · 2023-05-08
> > **Reply to Authors**
> >
> > The reviewer would like to thank the authors for their detailed response on the issues raised. The authors made a good job of explaining the merits of the search algorithm, such as searching for each stage, extraction block, and patch size and updating the paper accordingly. The authors made most of the changes pointed out by the reviewer. The visualization of the searched architecture gives the reader a good understanding of how the searched model looks and how it is different from the manually designed Patchcore.  Therefore, the reviewer is satisfied with the detailed response given by the authors and would like to update the rating to 7 (weak accept) from 5 (borderline learning reject).
> >
> > The reviewer is still skeptical regarding the original contribution of the search method. It is up to the Program Committee members to decide whether the application of NAS to a problem (visual anomaly segmentation in this case) is relevant to this conference.
> >
> > A few other minor issues: The references should not be referred to directly. For example, “in [12]” in line 298 should rather be written as Roth et al. [12]. A sentence should be able to read as if all the citations are removed. This style should be corrected throughout the paper.

---

### Official Review · Reviewer_jVhD · 2023-04-13

**Potential Impact On The Field Of Automl Rating:** 2
**Technical Quality And Correctness Rating:** 4
**Clarity Rating:** 3

**Summary Of Contributions:**

This paper proposes a novel application of NAS to visual anomaly segmentation called AutoPatch. AutoPatch uses nearest neighbor scoring, weight-sharing NAS, and weighted average precision to provide a pareto-optimal model for image segmentation. The main novelty introduced with AutoPatch is the use of a NAS algorithm specifically tailored for visual anomaly segmentation. AutoPatch appears to improve performance in a multi-objective space, and is able to accurately identify a near-optimal neural architecture for that problem with only a few examples of the anomaly.

**Actions Required To Increase Overall Recommendation:**

I would recommend re-organizing the paper. Moving Figure 2 to the appendix, and moving the appendix after the citations would help improve the readability of the paper.

**Clarity:**

This paper was generally written in a very clear and concise manner. The organization of the paper led to a small amount of confusion. Section 5.1 was also a bit confusing to follow in bullet points. Perhaps using a different layout for the information, such as a table, would be beneficial.

**Overall Review:**

Strengths:

The authors introduce a new method for visual anomaly segmentation, through the use of a NAS algorithm searching for subnetworks from MobileNetv3, in order to generate near pareto-optimal network architectures after being given only a few examples of a visual anomaly in a dataset. A novel weighted average precision metric is being used in this space in order to create a more robust performance indicator.

Weaknesses:

One minor weakness within the paper is the limited scope of experimentation. While there are good and potentially generalizable results shown on the MVTec dataset, this would be more substantial with further experimentation on other datasets.
This is partially addressed within the future works section along with another weakness I agree with. Synthesizing artificial anomalies could provide this angle for further experimentation. Also, further ablation studies on the generalization of the model on larger anomaly set sizes would improve the empirical strength of this paper.


**Potential Impact On The Field Of Automl:**

While currently limited in scope to the field of visual anomaly segmentation, this paper improves on prior works in a manner that allows for a more efficient and accurate solution for visual anomaly segmentation. Because of its limitations, this paper is not largely impactful for the general field of AutoML.

**Review Confidence:**

3: You are fairly confident in your assessment. It is possible that you did not understand some parts of the submission or that you are unfamiliar with some pieces of related work.

**Review Rating:**

7: Weak Accept: Technically sound paper with moderate-to-high impact and strong evaluation, with perhaps some minor flaws.

**Review Summary:**

This paper introduced a novel application for NAS, and a novel metric for image anomaly segmentation. It was presented in a clear manner with robust experimentation. There is a downside in its limitation to the field of visual anomaly segmentation, but this can be overcome with the usage of weighted average precision to address class imbalance.

**Technical Quality And Correctness:**

This paper showcased the performance of an anomaly segmentation algorithm using NAS on the MVTec dataset. There were also ablation studies run showcasing the performance of the model against the pareto frontier. Overall, the technical quality and correctness was good.

---

> ### Author Response · Authors · 2023-05-01
> **Response to Reviewer jVhD**
>
> Dear Reviewer jVhD,
>
> We sincerely appreciate your comprehensive review of our paper and your constructive feedback.
>
> **Regarding the Experimental Setup:** Your suggestion about refining the presentation in Section 5.1 is well received. We have replaced the bullet points with a table to better organize and communicate the information.
>
> **On the Use of Other Datasets:** We agree that diversifying the datasets used for testing would be beneficial. It's worth noting that we have treated MVTec as a meta-dataset, running separate experiments for each of its 15 distinct categories. This approach effectively means we've tested our method across 15 different datasets. We understand, however, the value of further diversifying the datasets used and will reserve this for future work.
>
> **Addressing Larger Anomaly Sets:** You raise a valid point about the potential value of larger anomaly sets. Our focus in this paper was to handle situations where very few anomalous examples are available, not enough for training a model via conventional supervision. Yet, we acknowledge the potential improvements larger anomaly sets could bring to the robustness of our method.
>
> **In Terms of Reorganizing the Paper:** We appreciate your suggestions on enhancing the readability of the paper. We have moved Figure 2 to the Appendix and relocated the Appendix itself to appear after the references as recommended.
>
> We believe that these revisions should address your concerns and further enhance the quality of our work.
>
> Thank you once again for your constructive feedback.

---

### Official Review · Reviewer_7MEw · 2023-04-13

**Potential Impact On The Field Of Automl Rating:** 2
**Technical Quality And Correctness Rating:** 3
**Clarity Rating:** 3

**Summary Of Contributions:**

This paper presents a framework called AutoPatch for segmenting visual anomalies. Essentially, it is a neural architecture search method that enables efficient segmentation of visual anomalies without any training. By leveraging a pre-trained supernet, a black-box optimization algorithm can directly minimize FLOPS and maximize wAP on a small validation set of anomalous examples. Finally, compelling results on the widely studied MVTec dataset are presented, demonstrating that AutoPatch outperforms the current state-of-the-art method.

**Actions Required To Increase Overall Recommendation:**

- discuss more about the proposed metric
- the method is very sensitive to the random seed ( please correct the reviewer if the understanding is wrong), which makes the proposed method less reliable

**Clarity:**

Overall, the presentation of this paper is clear and easy to follow. This paper proposed a new metric. However, this paper failed to discuss this metric in more detail, e.g. using more experiments, on more datasets, etc. I would suggest that authors spend more efforts discussing the its properties and evaluating it using more empirical experiments.

**Overall Review:**

Overall, the motivation of this paper is clear: try to balance performance and FLOPS for segmentation tasks using a weight-sharing NAS method.

strength:
- motivation is pretty clear and the problem this paper tries to solve is important.
- the presentation of the proposed method is clear

weakness:
- the method is a combination of weight-sharing NAS and NN score. Besides, the weight-sharing NAS part only involves inference, that is to say, this method takes a pre-trained weight-sharing supernet and directly uses it to do the multi-objective search. The novelty is limited.
- the evaluation is also limited. For example, a comparison with model compression technique is encouraging, although the result will be a single point in the Pareto front.

**Potential Impact On The Field Of Automl:**

This paper proposed to directly use a pre-trained supernet for segmentation and the anomaly detection method is based on nearest-neighbor score. The motivation is clear and this combination is to find a better trade-off between performance and FLOPS. However, the impact may not be large as the proposed method does not provide additional insights.

**Review Confidence:**

4: You are confident in your assessment, but not absolutely certain. It is unlikely, but not impossible, that you did not understand some parts of the submission or that you are unfamiliar with some pieces of related work.

**Review Rating:**

3: Reject: For instance, a paper with technical flaws, weak impact, and/or weak evaluation.

**Review Summary:**

Given the novelty and empirical evaluation is limited, the reviewer does not support acceptance of this paper.

**Technical Quality And Correctness:**

This paper is well-written and details are discussed extensively. The motivation of the proposed method is clear. Overall, the proposed approach lacks novelty to some extent.

---

> ### Author Response · Authors · 2023-05-01
> **Response to Reviewer 7MEw**
>
> Dear Reviewer 7MEw,
>
> We greatly appreciate the time and effort you put into reviewing our paper, and we value your insightful comments. We have carefully taken into account each point raised and have made revisions accordingly.
>
> **Regarding the Metric:** Your observation about the need for a more in-depth discussion of the newly introduced region-weighted Average Precision is well taken. We initially included a comparison with existing metrics in the Appendix, recognizing the importance of this, we have relocated this comparison to Section 5.4 in our revised paper, and have provided a more comprehensive analysis.
>
> **Addressing Novelty:** We acknowledge that our method primarily involves inference. The novelty, however, resides in the application of NAS to the visual anomaly segmentation problem. Specifically, we have innovated by using a pre-trained supernet, substantially reducing the computational load usually associated with NAS. This paper presents the first application of NAS to this demanding task. Additional innovative aspects include the distinctive approach of searching for each stage, extracting block and patch size, and setting the patch size per extraction block, rather than a singular patch size. Moreover, our dynamic depth feature enables further reduction of computational complexity. We also emphasize the value of our method's simplicity, as it can be effortlessly adapted and applied to real-world datasets.
>
> **Concerning Limited Evaluation:** We concur that the evaluation in the initial version of the paper was limited. To address this, we have broadened our experiments in the revised version. These include:
>
> - A comparison with PatchCore using a MobileNetV3-Large backbone (Section 5.3)
> - An evaluation utilizing different metrics (this content has been relocated from the Appendix to Section 5.4)
> - In-depth analysis of the architectures discovered by our method for each category (Section 5.5)
> - An ablation study on the search space (this content has been relocated from the Appendix to Section 5.6)
>
> **In terms of Seed Sensitivity:** We agree that our method displays sensitivity to the random seed. We have explicitly highlighted this as a limitation of our method in the revised paper and suggested for future work to minimize this sensitivity through augmentation and synthesis of additional anomalous examples. Additionally, we noted that the search method MOTPE exhibits a variance of one percentage point in the oracle evaluation setting, indicating that the search algorithm itself could potentially be improved. While this improvement is beyond the scope of this paper, we aim to address it in future work.
>
> We trust that these revisions adequately address your concerns and enhance the quality and impact of our work.
>
> Thank you once again for your valuable feedback.

---

### Comment · Program_Chairs · 2023-04-24
**Official reproducibility review**

The official comment "failed to install the required packages" is by the official reproducibility reviewer. (They could not submit the review as an official reproducibility review for technical reasons in OpenReview.) We would like to ask the authors to work with the reproducibility reviewer to resolve the installation issues to enable a proper reproducibility review. Thanks!